# Co-modulation of clinical courses of hypertension and depression in public health facilities in Ethiopia: A six-month follow-up study protocol

Anemaw Asrat[1]*, Getnet Mitike[2], Fentie Ambaw[1]

1 School of Public Health, College of Medicine and Health Sciences, Bahir Dar University, Bahir Dar, Ethiopia, 2 International Institute for Primary Health Care-Ethiopia, Addis Ababa, Ethiopia

* anemawasrat@gmail.com

## Abstract

### Background

The co-occurrence of hypertension and depression is increasing both in magnitude and complexity, negatively impacting treatment outcomes of both conditions. Given that both conditions are chronic, the impact of each condition over time on the other needs to be well understood. This project aims to examine how each condition affects the course and outcome of the other over a six-month follow-up period.

### Methods

Four hundred eighty-five adults with newly diagnosed hypertension will be screened for depression using the nine-item Patient Health Questionnaire (PHQ-9) at the time of diagnosis. We will employ a prospective cohort study design, with depression serving as the exposure variable. At baseline, blood pressure, depression level, co-morbid illness, perceived social support, substance use, pain level, health-related quality of life, and disability will be measured. At three and six months, blood pressure, depression level, health-related quality of life, and disability measurements will be repeated.

The prevalence of depression will be determined at baseline, and the factors associated with it will be examined using Poisson regression with a robust variance estimator. The incidence of depression will be calculated at 3 months and 6 months. The effect size of the change from baseline to endpoint will be measured using Cohen's d. To identify the predictors of change in PHQ-9 scores over time, multilevel mixed-effects linear regression will be employed. Blood pressure control at three and six months will be evaluated, and the impact of baseline depression on blood pressure control will be analysed. P-values less than 0.05 will be considered statistically significant.

**Data availability statement:** No datasets were generated or analysed during the current study. All relevant data from this study will be made available upon study completion.

**Funding:** This research is funded by Bahir Dar University under the BDU-LCM study grant number BDU-CMHS-005 The funders had no role in study design, data collection and analysis, decision to publish, or preparation of the manuscript.

**Competing interests:** The authors have declared that no competing interests exist.

## Ethics

The Institutional Review Board of the College of Medicine and Health Sciences, Bahir Dar University, has approved the study.

## Anticipated impact

The findings will contribute to the scarce body of knowledge about long-term interaction between hypertension and depression, particularly on the impact of unmanaged depression on hypertension control, and the incidence of depression in hypertension patients. Findings on the course of depression will help clinicians to decide on the dilemma of treating it or expecting spontaneous recovery with the effective management of hypertension.

## Introduction

Hypertension is a serious and common condition characterised by increased pressure in the blood vessels [1]. According to the 2021 Global Burden of Disease Study, hypertension accounted for 10.9 million deaths, or 16% of total deaths [2]. The same study documented that 225 million disability-adjusted life years (DALYs) were attributable to hypertension worldwide, making it the second leading cause, with 7.8% of the total DALYs [3].

Hypertension is the most significant modifiable risk factor for cardiovascular diseases (CVD) [4,5], which are the leading causes of premature morbidity and mortality globally [4,5]. The number of adults aged 30–79 years with hypertension has increased from 650 million to 1.28 billion in the last thirty years, and its trend shows a clear shift in burden from high-income to low-income regions [6].

Better hypertension management significantly saves lives. For example, increasing the percentage of people whose hypertension is under control worldwide to 50% could prevent 76 million deaths from 2023 to 2050 [7]. Unfortunately, only 54% of adults aged 30–79 years with hypertension are being diagnosed, 42% are receiving treatment for their hypertension, and 21% have had their condition controlled [7]. Treating hypertension is one of the most crucial actions to achieve the Sustainable Development Goal (SDG) target 3.4, which aims for a one-third reduction in premature deaths from the leading noncommunicable diseases [7]. However, according to the 2022 World Health Organisation's non-communicable diseases progress monitor, most countries of the world are significantly lagging behind the targets of sustainable development in reducing chronic non-communicable diseases, as a result of which they are still responsible for 74% of deaths [8].

The co-occurrence of hypertension and depression is increasing, affecting treatment outcomes of both conditions negatively. The prevalence of depression in people with hypertension is 26.8% globally [9], and in the range of 26.7% to 37.8% in Ethiopia [10–13]. The impact of comorbidity on a patient's overall well-being is greater than the sum of each condition's individual effect [14]. Individuals with depressive

symptoms are likely to develop hypertension, and vice versa, indicating a bidirectional relationship. The use of antidepressant medications is also known to interfere with blood pressure control in patients with hypertension [15].

However, a number of questions that are critical to patient care and further understanding of the course of the two chronic conditions remain unaddressed. Given the problem of introducing antidepressant medications to hypertension patients, whether merely managing hypertension resolves the depression or not is an important consideration; however, no evidence exists either to support or reject the idea. National guidelines for hypertension management do not include a protocol for comorbid depression management [16]. In relation to this, the impact of untreated depression on the course and outcomes of hypertension, including blood pressure control, quality of life, and disability, is not well understood. Furthermore, the development of new depression cases and relevant factors associated with it in hypertension patients who are already under the attention of healthcare providers within the health system is not known. This is a critical concern because the management of hypertension requires the full engagement of patients, and those who develop depression will have compromised self-care. Addressing this final question may help to improve the quality of services to patients on follow-up care. The overall aim of this study is to investigate the course and outcomes of hypertension and depression when they occur together in health facility settings in Ethiopia. Our specific objectives are to: determine the prevalence and associated factors of depression at the time of hypertension diagnosis in adults; determine the incidence of depression at three and six months after a hypertension diagnosis in adults; examine the impact of hypertension on the course and outcome of depression at three and six months after diagnosis; and, examine the impact of untreated depression on the course and outcome of hypertension at three and six months after diagnosis.

## Materials and methods

### Study setting

The study will be conducted in public health facilities of Bahir Dar City from September 15, 2025, to June 30, 2026. We anticipate having the study results by August 2026. Bahir Dar is the capital city of the Amhara regional state and has a population of more than 750,991 residents [17]. There are three public hospitals and six health centers in the city. Based on a minimum of 5 patients/ month recruitment potential criteria, three hospitals and four health centers providing NCD care have been selected.

Hypertensive patients in the study health facilities are managed according to the Ethiopian National Noncommunicable Diseases Management Protocol [16]. According to this protocol, all adults visiting health facilities have their blood pressure measured as part of vital signs. The diagnosis of hypertension is confirmed by trained nurses with a BSc degree or Public Health Officers in health centres or by General Practitioners (medical doctors) in hospitals. In a health care facility with electricity or regular access to batteries, the Ministry of Health-Ethiopia recommends using an automated, validated blood pressure device with a digital reading to measure blood pressure. If the health care facility has no electricity or batteries, a manual (Aneroid or Mercury) BP apparatus can be used by auscultation with a stethoscope.

Once the patients' diagnosis is confirmed, they will be linked to a follow-up clinic typically staffed with general practitioners, nurses, or public health officers. Under normal circumstances, patients attend the clinic monthly to refill their medications, adjust medication, and achieve the target blood pressure.

### Study design

The study will employ a prospective cohort design in which adults with newly diagnosed hypertension will be recruited at the time of enrolment to care and followed for 6 months. Data will be collected at three time points: baseline, 3 months, and 6 months. The study flow is clarified more in the study framework (Fig 1).

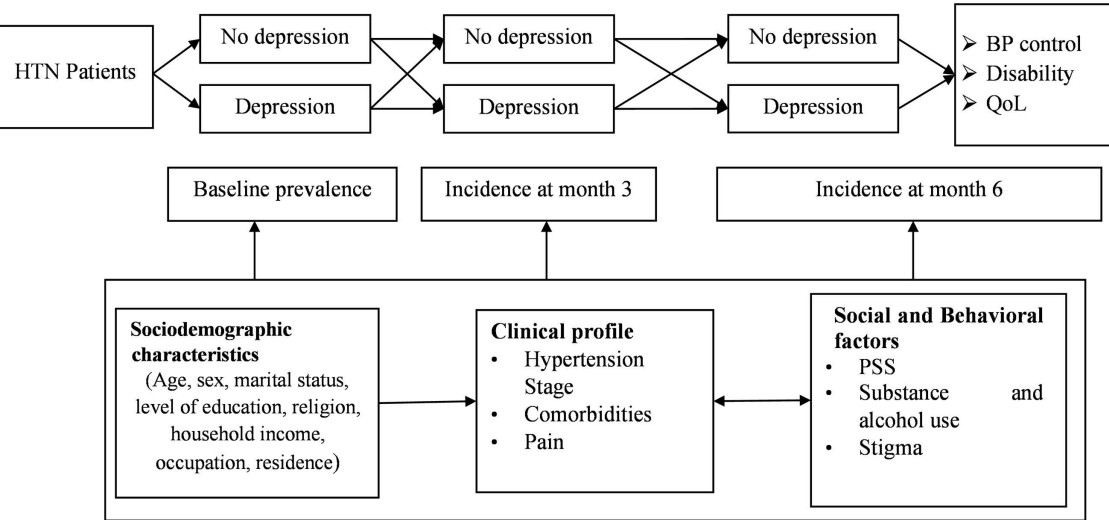

**Fig 1. study framework adapted from Ambaw and colleagues [18].** *HTN = Hypertension; BP = blood pressure; QoL = quality of life; PSS = perceived social support.*

## Recruitment and data collection

The health professionals running the NCD clinic at each site will identify the study participants who meet the inclusion criteria consecutively until a total of 485 participants is reached, and link them to experienced and trained data collectors, who are BSc degree holders in health. The health professionals in the NCD clinics will also play a role in helping the data collectors review patients' charts for hypertension treatment outcomes and comorbid illnesses. The data collectors will provide the patient with information about the study and seek their informed consent. They will enrol patients who consent and arrange appointment dates for the subsequent interviews, and carry out interviews at the health institutes.

## Eligibility criteria

**Inclusion criteria.** Newly diagnosed adult hypertension patients will be included in the study.
**Exclusion criteria.** The following groups of patients will be excluded.

- Patients who plan to be transferred out of Bahir Dar will be excluded because it will be difficult to take second-time and third-time measurements.

- Patients with verbal communication difficulties will be excluded because of the absence of a validated tool and experts for sign language. However, we do not anticipate to have such cases in significant numbers.

Seriously ill patients who cannot provide baseline data because of their illness
Patients who require admission at the time of diagnosis will have additional psychosocial stressors that will not be measured or analysed in the study.

## Sample size determination

The sample size has been calculated using the "Cohort or Cross-sectional study" sub-menu of Epi Info version 7.2. 1 (https://www.cdc.gov/epiinfo/index.html) with a 95% confidence level and 80% power. We used the proportion of ' unsuccessful treatment outcomes' in hypertension patients *with baseline depression* and *without baseline depression* to

be 12.9% and 3.4% respectively, from a proxy in TB patients [19]. The ratio of 3:1 for unexposed to exposed, along with a 20% contingency for loss to follow-up, was considered. The required sample size is 404. The final target sample size, including the contingency, is 485 (363 without depression and 122 with depression).

## Sampling technique

We will use consecutive sampling, where participants will be included in the study as long as they fulfil the inclusion and exclusion criteria until the targeted sample size is reached. Consecutive sampling technique is a common practice in healthcare research, particularly when the study population is infinite. It is claimed to be reasonably representative when: a) multiple sites are included, b) all eligible participants are invited to participate, and c) the data collection period is long [20]. The participants of this study will be recruited from seven centres over ten months

## Variables and measurement

**The exposure variable. Depression** is the exposure variable. It will be measured using the nine-item version Patient Health Questionnaire (PHQ-9). This scale has been widely used in surveys, effectiveness trials, and cohort studies in various populations [21]. In Ethiopia, PHQ-9 has been validated twice and was useful in screening depression in adult outpatients in both studies [22,23]. Given the potential overlap between hypertension and depression symptoms, the more conservative cut point of ten and above will be applied to define depression in our sample.

**Outcome variables.** *Primary outcomes*. Blood pressure control status is the primary outcome of the study. It will be categorized as controlled (systolic blood pressure of <140 mm Hg and diastolic blood pressure of <90 mm Hg) and not controlled otherwise. Baseline values will be recorded at the time of diagnosis. Follow-up assessments will be done at three and six months. Measurement procedures will be according to the Ethiopian National guideline [16]. Participants who drank caffeinated beverages (tea or coffee) or had been working within 30 minutes would wait 30 minutes before blood pressure measurements. The assessors will make the participants sit with their backs straight and supported on a chair or wall, their feet flat on the floor, and their legs uncrossed with their upper arm at heart level. The first measurement will be taken after resting for at least 5 minutes, and the second will be measured 5 minutes after the first measurement. The average of the two measurements will be taken to define blood pressure control.

*Secondary outcomes*. Secondary outcomes include disability and health-related quality of life (QoL). Secondary outcomes will be assessed at baseline, as well as at three and six months.

i. Disability

Disability will be measured using the interviewer-administered version of the 12-item World Health Organisation Disability Assessment Schedule, version 2.0 (WHODAS 2.0) [24]. Studies have demonstrated the usefulness of the tool for assessing disability in primary care patients with depression [25] and its capacity to capture changes [26]. WHODAS has been adapted for Ethiopia and has been shown to have good convergent validity in people with depression [27]. The "simple scoring" approach will be used, where the points assigned to each response category ("none"= (1), "mild" = (2), "moderate" = (3), "severe"= (4), and "extreme" = (5)) are summed to compute total scores. Higher scores will be indicative of a greater burden of disability.

ii.   Health-related quality of life (QoL)

QoL will be measured using a single-item "How would you rate your health-related quality of life?" with responses quantified using a numerical scale ranging from zero (the worst imaginable QoL) to ten (the highest imaginable QoL) [28]. Single-item methods have been used successfully in population surveys, clinical settings, and clinical interviews [29] and found to be valid in showing vulnerability to all-cause mortality [30]. In the study area, the scale was successfully used in a cohort of tuberculosis patients [19].

## Independent variables

Independent variables will be measured using previously successfully applied tools.

*Socio-demographic variables:* Data will be collected on age, sex, marital status, level of education, religion, household income, occupation, and place of residence (urban versus rural).

*Pain:* Pain will be measured using a visual analogue scale, where values will be scored from 0 (absence of symptom) to 10 (worst level of symptom severity), as perceived by the respondent for that week. The responses will be categorised as follows: no symptom (0), mild (12–3), moderate (4–6), and severe (7–10) to indicate intensity [31].

*Substance use*: Alcohol, tobacco products, and khat use will be assessed using the WHO Alcohol, Smoking and Substance Involvement Screening Test, version 3.1 (ASSIST 3.1) [32]. ASSIST is a valid and reliable scale for use in cross-cultural settings, to assess multiple drug use, to detect different levels of drug use, and is applicable in primary care settings [33–36]. In this study, the various types of tobacco and khat will be considered similar. Similarly, the local alcoholic drinks 'tella,' 'tej,' and 'araki' will be considered standard drinks. The alcoholic percentages per volume for these Ethiopian traditional alcoholic beverages are 3.84 to 6.48, 8.94 to 13.16 and 33.95 to 39.9% respectively [37].

*Co-morbid illness:* Data on the presence of chronic illnesses other than hypertension will be collected by asking the respondents if they have been diagnosed with any chronic illnesses.

*Perceived social support* (PSS) is the expectation that support will be provided from one's social network [38]. PSS will be measured using the three-item Oslo Scale of Perceived Social Support (Oslo 3) [39] at continuous levels, with higher scores showing better perceived social support.

**Severity of hypertension:** Severity of hypertension will be categorised into five grades according to the national guideline as Grade-1, Grade-2, Grade-3, 'Hypertension Emergency', and 'Hypertension Urgency'.

## Method of analysis

Descriptive statistics will be used to summarise both dependent and independent variables. The prevalence of depression among patients at baseline will be determined by calculating the proportion of patients scoring 10 or more on the PHQ-9 scale. Factors associated with depression at baseline will be analysed using Poisson regression with a robust variance estimator, a recommended approach for analysing prevalence ratios in cross-sectional studies with binary outcomes where the outcome is common [40,41,42]. The proportion of patients with hypertension who scored below 10 on PHQ-9 at baseline but had scores of 10 or above at the end of the third month will be calculated to determine the incidence (risk) of depression at 3 months. This will be repeated at 6 months. To examine the course of co-morbid depression, the percentage of participants with over 50% reduction in PHQ-9 scores will be calculated at three and six months. The effect size of the change from baseline to endpoint will be measured using Cohen's d with Hedges' correction after checking correlation coefficients to be around 0.5 and standard deviations to become equal, as recommended by Lakens [43]. To identify the predictors of reduction or change in PHQ-9 scores over time, multilevel mixed-effects linear regression with maximum likelihood estimation will be employed. Repeated measurements will be considered nested within individual participants, and individuals will be nested within health institutions. Assessment times will be coded as 0, 3, and 6 for baseline, 3-month, and 6-month assessments, respectively.

Blood pressure control status at three and six months will be classified as controlled or not controlled. To evaluate the impact of baseline depression on hypertension control, risk ratios between participants with and without baseline depression will be calculated using Poisson working model with a robust variance estimator. Follow-up time will be incorporated as a weighting variable in the analysis of each outcome.

To examine the impact of co-morbid depression on disability and quality of life, an independent samples t-test will be used. To assess changes in health-related quality of life and disability scores from the time of diagnosis to the end of

follow-up, a multilevel mixed-effects generalised linear model will be fitted, considering the three measurement times (baseline, three months, and six months) as nested within individuals, and individuals nested within health institutions. Time will be centred at baseline, with baseline coded as zero, the third month as 3, and six months as 6.

All our inferential analyses, including independent variables in the multivariable analysis, will be based on a) their theoretical importance and b) the adequacy of the number of cell participants for each category. P-values less than 0.05 will be considered statistically significant.

## Ethical considerations

Ethical approval was obtained from the Institutional Review Board (IRB) of the College of Medicine and Health Sciences, Bahir Dar University, on April 2, 2025 (Protocol number: 3095/2025). Written informed consent will be obtained from participants. A data collector will read the information sheet and the consent form for participants who cannot read or write; they will then sign in the presence of a witness. All participants who endorse the suicide item or score ten or above on PHQ-9 will be referred to staff at the health institutions for further psychiatric evaluation. Participants will be free to withdraw from the study at any time.

## Acknowledgments

We are grateful to the BDU-LCM research team for their unreserved technical advice. We are sincerely grateful to Dr. Biset Ayalew Nigatu, an expert in the English language at Bahir Dar University, for editing the final manuscript.

## Author contributions

**Conceptualization:** Anemaw Asrat, Fentie Ambaw.

**Writing – original draft:** Anemaw Asrat, Fentie Ambaw.

**Writing – review & editing:** Anemaw Asrat, Getnet Mitike, Fentie Ambaw.

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
