## [Decision Letter · Decision Letter 0]

6 Jan 2026

Dear Dr.Anemaw Asrat,

Thank you for submitting your manuscript to PLOS ONE. After careful consideration, we feel that it has merit but does not fully meet PLOS ONE’s publication criteria as it currently stands. Therefore, we invite you to submit a revised version of the manuscript that addresses the points raised during the review process.

We look forward to receiving your revised manuscript.

Kind regards,

Nitai Roy

Academic Editor

PLOS One

Journal Requirements:

“This research is funded by Bahir Dar University under the BDU-LCM study grant number BDU-CMHS-005”

Reviewers' comments:

Reviewer's Responses to Questions

**Comments to the Author**

1. Does the manuscript provide a valid rationale for the proposed study, with clearly identified and justified research questions?

Reviewer #1: Yes

2. Is the protocol technically sound and planned in a manner that will lead to a meaningful outcome and allow testing the stated hypotheses?

Reviewer #1: Yes

3. Is the methodology feasible and described in sufficient detail to allow the work to be replicable?

Reviewer #1: No

4. Have the authors described where all data underlying the findings will be made available when the study is complete?

Reviewer #1: Yes

5. Is the manuscript presented in an intelligible fashion and written in standard English?

Reviewer #1: Yes

You may also provide optional suggestions and comments to authors that they might find helpful in planning their study.

Reviewer #1: Thank you for giving the chance to review a manuscript entitled “Co-modulation of clinical courses of hypertension and depression in public health facilities in Ethiopia: a six-month follow-up study protocol”. I want to congratulate the researchers for their planning. To increase the quality of the protocol, I have the following concerns.

In recruitment and data collection on page 7, line 81 & 82 “the study participants 82 who meet the inclusion criteria consecutively until a total of 485 participants (122 cases) is reached”. Here, your sample size is 485 (122 cases), which is manageable and it is expected that there will be huge number of such study participants in Bahir Dar city. If so, why the researchers planned and preferred to use “consecutively” than any random methods? Alternatively, why not systematic random sampling method?

The researchers planned to include newly diagnosed adult hypertension patients and evaluate after 3 & 6 months. Is depression detectable at 3 month duration of hypertension? Will the level of co-morbidity be significant at this level? To become with best research finding, better to extend the study duration and the evaluation months as depression takes time.

The inclusion criteria focused on newly diagnosed adult hypertension patients. However, your consideration under is sample size is with depression 485 (363 without depression and 122 with depression). This raises a question “can we gain depression cases from newly diagnosed hypertensive patients?” Please, consider this.

Thank you again in advance.

**Do you want your identity to be public for this peer review?** For information about this choice, including consent withdrawal, please see our Privacy Policy

Reviewer #1: **Yes:** Elias Ezo

---

## [Author Response · Author response to Decision Letter 1]

18 Jan 2026

This research is funded by Bahir Dar University under the BDU-LCM study grant number BDU-CMHS-005. The funders had no role in study design, data collection and analysis, decision to publish, or preparation of the manuscript.

---

## [Decision Letter · Decision Letter 1]

12 Feb 2026

Co-modulation of clinical courses of hypertension and depression in public health facilities in Ethiopia: a six-month follow-up study protocol

PONE-D-25-46537R1

Dear Dr. Anemaw Asrat%,

We’re pleased to inform you that your manuscript has been judged scientifically suitable for publication and will be formally accepted for publication once it meets all outstanding technical requirements.

Kind regards,

Nitai Roy

Academic Editor

PLOS One

Additional Editor Comments (optional):

Reviewers' comments:

Reviewer's Responses to Questions

**Comments to the Author**

1. Does the manuscript provide a valid rationale for the proposed study, with clearly identified and justified research questions?

Reviewer #1: Yes

2. Is the protocol technically sound and planned in a manner that will lead to a meaningful outcome and allow testing the stated hypotheses?

Reviewer #1: Yes

3. Is the methodology feasible and described in sufficient detail to allow the work to be replicable?

Reviewer #1: Yes

4. Have the authors described where all data underlying the findings will be made available when the study is complete?

Reviewer #1: Yes

5. Is the manuscript presented in an intelligible fashion and written in standard English?

Reviewer #1: Yes

You may also provide optional suggestions and comments to authors that they might find helpful in planning their study.

Reviewer #1: The authors addressed all the important parts raised in the previous comments and reacted well. The protocol is currently in acceptable level.

**Do you want your identity to be public for this peer review?** For information about this choice, including consent withdrawal, please see our Privacy Policy

Reviewer #1: **Yes:** Elias Ezo Ereta

---

## [Editor Report · Acceptance letter]

PONE-D-25-46537R1

PLOS One

Dear Dr. Asrat,

I'm pleased to inform you that your manuscript has been deemed suitable for publication in PLOS One. Congratulations! Your manuscript is now being handed over to our production team.

Kind regards,

on behalf of

Dr. Nitai Roy

Academic Editor

PLOS One